# Experimental and Numerical Study on the Dynamic and Flow Characteristics of a Reciprocating Pump Valve

**Ran Li** [1,2] , **Wenshu Wei** [2] , **Hao Liu** [2,*], **Jian Ye** [2], **Dalong Wang** [3], **Shoubin Li** [2] and **Wei Wang** [2]

1   Graduate School, China Coal Research Institute, Beijing 100013, China; liran@tdmarco.com
2   Beijing Tianma Intelligent Control Technology Co., Ltd., Beijing 100013, China; wws@tdmarco.com (W.W.); yejian@tdmarco.com (J.Y.); shoubin@tdmarco.com (S.L.); wangwei@tdmarco.com (W.W.)
3   School of Mechanical Electronic and Information Engineering, China University of Mining and Technology Beijing, Beijing 100083, China; wangdl@tdmarco.com
*   Correspondence: liuhao@tdmarco.com

**Abstract:** The structure and dynamics of a reciprocating pump liquid end affect the volumetric efficiency and net positive suction head. To match the kinematics with theoretical parameters, reciprocating pump valve motion and flow visualization tests and computational fluid dynamics (CFD) analyses were performed on a wing-guided bevel discharge valve in a horizontal quintuple single-acting reciprocating pump. The valve motion test results showed that the maximum pump valve displacement and the pump valve opening and closing durations were approximately 8.3 mm, 29 ms, and 38 ms, respectively. The corresponding flow visualization test results were 11.4 mm, 9.5 ms, and 35.5 ms. The valve closing durations obtained from the valve motion and flow visualization tests are approximately twice as high as the U-Adolph prediction. The maximum displacement obtained from the valve motion test is consistent with the U-Adolph prediction. Three-dimensional CFD analyses were performed to investigate the flow states, pressure, and velocity characteristics of the discharge valve opening. Finally, the proposed method was applied to develop a new horizontal quintuple single-acting reciprocating pump with a rated flow rate of 1250 L/min and pressure of 40 MPa. This developed pump exhibited good performance and excellent reliability.

**Keywords:** reciprocating pump; valve motion; visualization test; LVDT; high-speed camera; U-Adolph theory; CFD

## 1. Introduction

Reciprocating pumps are widely used in petroleum/gas, chemical, mining, and other industrial applications [1,2]. Self-acting pump valves at the inlet and outlet of the liquid end of a reciprocating pump are used for breathing and fluid removal. The dynamic and flow characteristics of reciprocating pump valves are often associated with volumetric efficiency and valve failures, such as wear and cavitation. Therefore, it is necessary to ensure that the dynamic and flow characteristics of reciprocating pump valves are well understood. Experimental measurements are vital for describing the flow and dynamic characteristics and are widely used to modify analytical and numerical models. Sensors, such as position, velocity, acceleration, and force sensors, are essential instruments for measuring the dynamic parameters of valves [3–5]. In many applications, high-speed cameras are typically used to optically detect flow characteristics, such as bubbles and cavitation [6–9]. Computational fluid dynamics (CFD) analyses are commonly used to study the flow characteristics of pump valves [10–12].

Analytical, numerical, and experimental studies have been conducted to investigate the dynamic and flow characteristics of reciprocating pump valves. U-Adolph proposed a second-order nonlinear differential equation for describing the pump valve movement [13]. Because the U-Adolph equation results in a singular solution when used to describe valve opening and closing, it is only suitable for describing the valve motion after the

valve opening. Johnston [14] developed a numerical model considering cavitation for predicting the dynamic behavior of a reciprocating pump with self-acting valves. Opitz and Schlücker [15] investigated the effects of cavitation in a reciprocating pump using a high-speed camera. Pei et al. [16,17] experimentally analyzed the characteristics of the valve motion of a reciprocating pump based on acceleration sensors and studied the collision contact characteristics of a reciprocating pump via numerical analysis using the commercial code ANSYS/LS-DYNA. Wang et al. [18] investigated the effects of spring stiffness and valve quality on the motion characteristics of a reciprocating plunger pump using fluid–structure interaction (FSI) and experimental methods. Aldo et al. [19] conducted a CFD study with experimental tests to investigate cavitation in a positive displacement pump during the suction stroke. Dong et al. [10] performed three-dimensional (3D) transient analyses to investigate the change rule of the suction coefficient during the suction valve opening of a reciprocating pump. Li et al. [11,20] studied cellular automata techniques and numerical simulations based on the commercial codes AMESim and ANSYS-FLUENT to investigate the dynamic and flow characteristics of the valves in emulsion reciprocating pumps. Ma et al. [12] conducted CFD numerical analyses and particle image velocimetry (PIV) visualization experimental studies to investigate the two-phase flow fields of three typical valves in reciprocating multiphase pumps. Li et al. [21] studied the influence of the flow distribution structure on the output flow pulsations of an axial piston pump using Matlab in conjunction with the AMESim commercial code. Yang et al. [22] used the CFD method to study the influence of the leading edge of a centrifugal pump blade and impeller fillet on the overall performance. As a comprehensive study on the dynamic and flow characteristics of a reciprocating valve, in terms of the analytical model, CFD simulations, and valve motion and flow visualization tests, has not been reported to date, this study aims at conducting such an investigation via experiments and simulations.

This study introduces a valve motion and flow visualization test program and performs an experimental investigation. The valve motion test was performed using a data acquisition system in conjunction with a linear variable differential transformer (LVDT) sensor to characterize the dynamic behavior of the discharge valve of a horizontal quintuple single-acting reciprocating pump. The flow visualization test was conducted using a high-speed camera to investigate the flow characteristics during the valve opening and closing processes. The experimental results were compared with numerical predictions based on the U-Adolph theory. The maximum displacement of the pump valve opening, obtained from the valve opening, and the duration of valve opening and closing, obtained from the flow visualization test, were consistent with the U-Adolph predictions. A series of 3D CFD analyses were performed at different valve-opening lifts. The flow states, pressure, and velocity characteristics during discharge valve opening were numerically obtained. In contrast to the visualization test results, no bubbles were observed in the CFD results. The influence of the valve-opening lift on the discharge coefficient and velocity is discussed. Additionally, the proposed method was used to develop a new horizontal quintuple single-acting reciprocating pump, whose application is also presented herein.

## 2. Valve Motion and Flow Visualization Tests

### 2.1. Valve Motion Test

A BRW630/40 horizontal quintuple single-acting reciprocating pump was used in this study. The test pump was driven by an electric motor and breathed the water medium through a water tank and a piping system. The water medium was removed through a pressure-relief valve, from which the water medium was returned to the water tank. The operating pressure was approximately 37.5 MPa.

A Soway SDVG20 LVDT sensor, with a measurement range of 0–15 mm and a resolution within 2% f.s., was mechanically coupled to the pump valve, as shown in Figure 1. The SDVG20 LVDT sensor converted the rectilinear motion of the pump valve to a corresponding current signal. Figure 2 shows the testing system for pump valve motion. The output current signal measured using the SDVG20 LVDT sensor was sent to a National Instrument

(NI) 9203 signal conditioning module for test data postprocessing. A real-time digital filter algorithm was used to process the current signals.

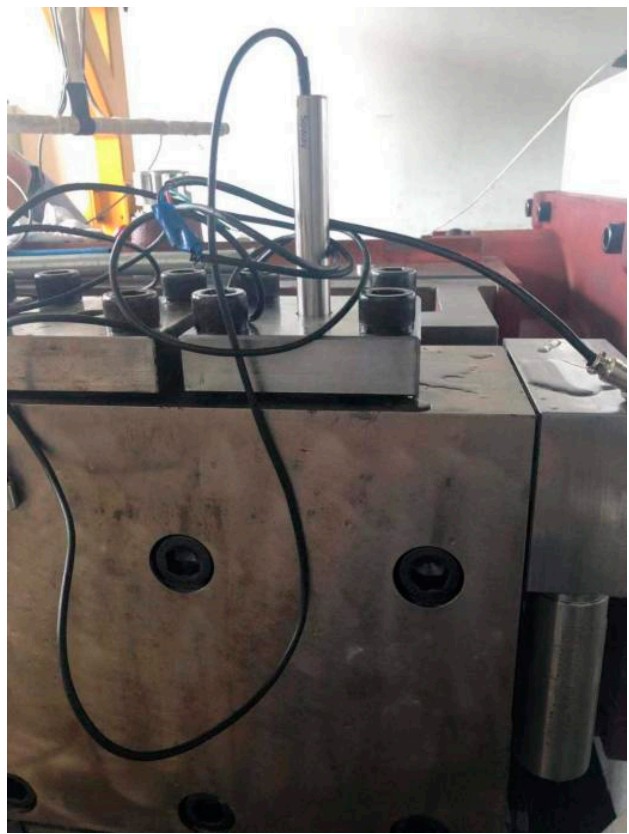

**Figure 1.** Setup of the LVDT sensor for monitoring the displacement of the pump valve.

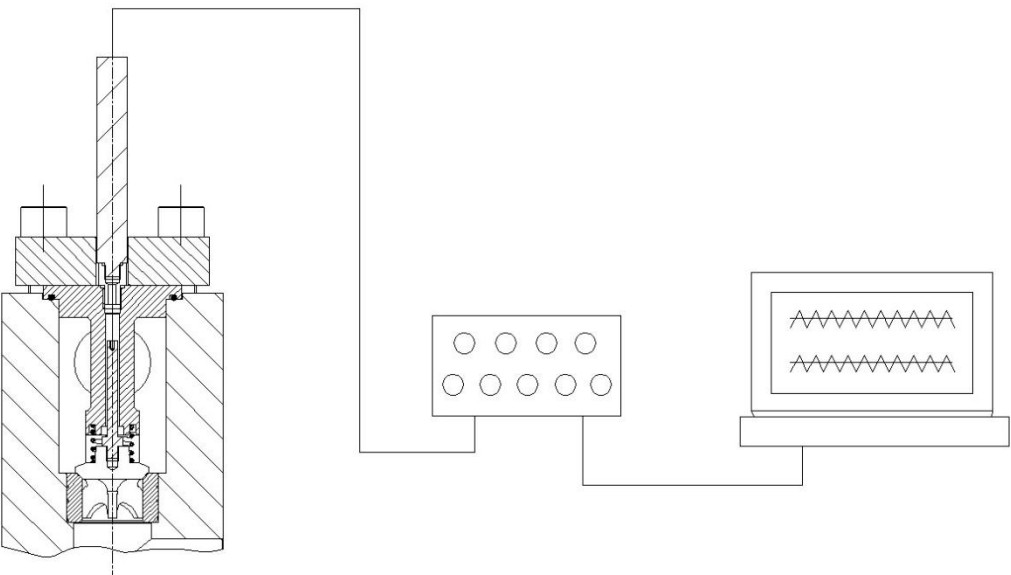

**Figure 2.** Schematic of the pump valve motion testing system.

A wing-guided bevel discharge valve was tested. The straight interference fit retention method was used for the valve seat. A 45° bevel angle ($\theta$) was designed to induce better centering during valve closing and decrease flow turbulence. The valve body mass $m$ was 0.425 kg, and the spring stiffness coefficient $c$ was 6.842 N/mm. The outer diameter of the

valve $D_{outer}$ and inner diameter of the valve seat orifice $D_{inner}$ were 48 mm and 40 mm, respectively. The pump valve opening and closing frequency is related to the crankshaft speed and plunger stroke. The crankshaft speed and plunger stroke were 663 rpm and 70 mm, respectively. The test pump and pump valve parameters are summarized in Table 1.

**Table 1.** Test pump and pump valve parameters.

| Tested Specimen | Parameter | Numerical Value |
|---|---|---|
| Pump | Flow rate (L/min) | 630 |
| | Crankshaft speed (rpm) | 663 |
| | Stroke (mm) | 70 |
| | Tested operating pressure (MPa) | 35 |
| Discharge valve | Mass of discharge valve (kg) | 0.26 |
| | Spring stiffness (N/mm) | 3.8 |
| | Suction valve spring preload (N) | 38.5 |
| | Diameter of discharge valve (mm) | 48 |
| | Effective flow diameter of discharge valve (mm) | 40 |

The displacement obtained from the LVDT sensor is plotted against time, as shown in Figure 3. The maximum displacement of the pump valve was approximately 8.3 mm. The valve motion test data showed that the opening and closing durations of the pump valve were approximately 29 ms and 38 ms, respectively.

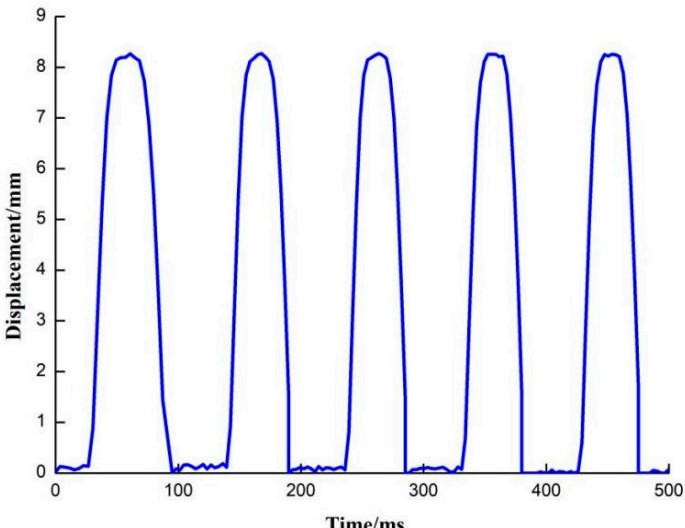

**Figure 3.** The valve displacement curve obtained from the valve motion test.

### 2.2. Flow Visualization Test

A prototype liquid end made of plexiglass with a one wing-guided bevel valve and one valve seat was connected to the power end of the BRW630/40 reciprocating pump. The wing-guided bevel valve was the same as that used in the valve-motion test. Owing to the brittleness of plexiglass, the radial O-ring seal retention method was used instead of the interference fit retention method. Compared with the valve motion test, the operating pressure was reduced to 1.5 MPa, considering the strength of the plexiglass. A high-speed camera (Phantom VEO 710) was used to visualize the flow characteristics during the valve opening and closing. The Phantom VEO 710 high-speed camera is capable of capturing images at 7Gpx/s for frame rates of up to 7500 frames per second (fps) at a 1280 × 800 resolution and a minimum exposure time of 1 μs. In this study, a resolution of 832 × 600 pixels with a frame rate of 10,000 fps and an exposure time of 98.75 μs was used. Two additional lights were used to assist the visualization. The schematic and test apparatus for

the pump valve visualization test are shown in Figure 4. The video was recorded on an in-situ memory card, which was used for data postprocessing.

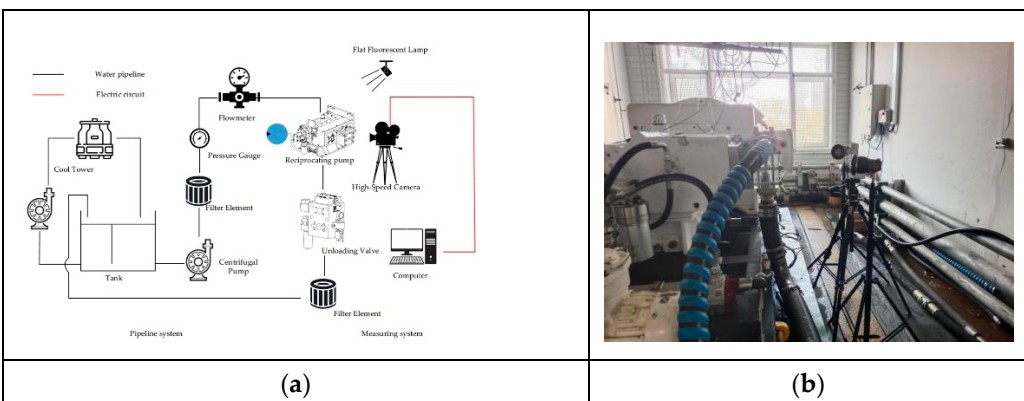

|          |          |
|:--------:|:--------:|
|   (**a**)   |   (**b**)   |

**Figure 4.** Pump valve visualization test: (**a**) schematic of the experimental setup; (**b**) photograph of the experimental facility.

The Phantom CineView software was used to perform the measurements and apply image processing. The software allows a grid display, as shown in Figure 5. This assists in converting the coordinate information into the movement of the pump valve. The batch processing function of the software was implemented for image postprocessing. The dynamic and flow characteristics during the valve opening and closing processes were interpreted from these images. The maximum displacement and valve opening and closing durations were approximately 11.4 mm, 9.5 ms, and 35.5 ms, respectively.

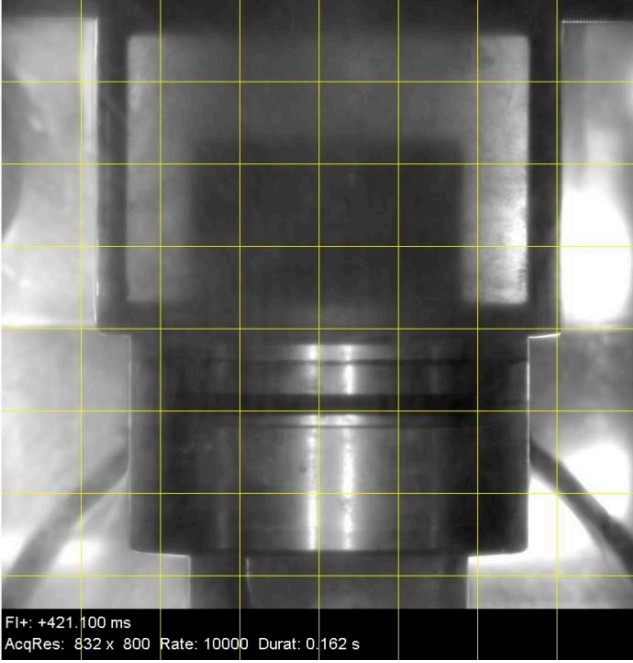

**Figure 5.** Grid image analysis method.

## 3. Numerical Analysis of Valve Opening Dynamics and Flow Characteristics
*Three-Dimensional CFD Model of the Pump Valve*

The commercial code ANSYS-FLUENT [23] was used to model the discharge valve opening in this study. The simplified model was built by taking the discharge valve as a rigid body with varying opening lifts, $h$, ranging from 1 to 7 mm. A typical model consisting of the fluid domain and the boundary and magnified view of the discharge

valve mesh is shown in Figures 6a and 6b, respectively. The standard k–ε turbulence model used in this study is a typical two-equation model, which has been proven to have relatively good stability, reasonable accuracy, and fast convergence in valve and orifice flow simulations [24]. The boundary conditions used in the simulation were a constant velocity of 1.3 m/s (at the inlet section) and a constant pressure of 37.5 MPa (at the outlet section of the discharge valve). The medium was pure water with a density of 1000 kg/m$^3$.

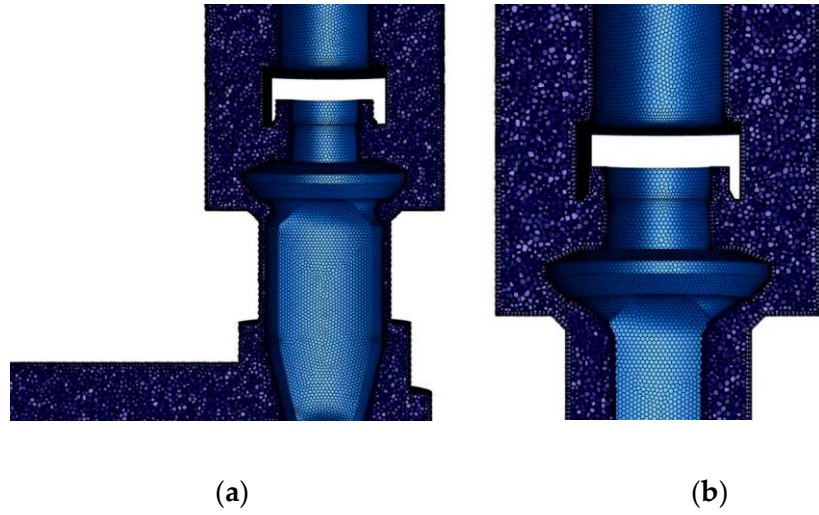

(**a**)                                  (**b**)

**Figure 6.** (**a**) Meshed fluid domain; (**b**) magnified view of the mesh.

The continuity, momentum, and standard $k$–$\varepsilon$ model equations are expressed in the Cartesian coordinate system in Equations (1)–(4).

$$\frac{\partial \rho}{\partial t} + \frac{\partial (\rho u_i)}{\partial x} = 0 \tag{1}$$

$$\frac{\partial \rho u_i}{\partial t} + \frac{\partial \rho u_i u_j}{\partial x_j} = -\frac{\partial P}{\partial x_i} + \left[ (\mu + \mu_t)\left( \frac{\partial u_i}{\partial x_i} + \frac{\partial u_j}{\partial x_i} - \frac{2}{3}\delta_{ij}\frac{\partial u_i}{\partial x_j} \right) \right] \tag{2}$$

$$\frac{\partial}{\partial t}(\rho k) + \frac{\partial}{\partial x_i}(\rho k u_i) = \frac{\partial}{\partial x_j}\left[ \left( \mu + \frac{\mu_t}{\sigma_k} \right)\frac{\partial k}{\partial x_j} \right] + G_K + G_b - \rho\varepsilon - Y_M + S_k \tag{3}$$

$$\frac{\partial}{\partial t}(\rho\varepsilon) + \frac{\partial}{\partial x_i}(\rho\varepsilon u_i) = \frac{\partial}{\partial x_j}\left[ \left( \mu + \frac{\mu_t}{\sigma_\varepsilon} \right)\frac{\partial k}{\partial x_j} \right] + G_{1\varepsilon}\frac{\varepsilon}{k}(G_K + G_{3\varepsilon}G_b) - G_{2\varepsilon}\rho\frac{\varepsilon^2}{k} + S_\varepsilon \tag{4}$$

where $\delta_{ij}$ is the Kronecker symbol, $G_k$ and $G_b$ are the generations of turbulent kinetic energy due to the mean velocity gradients and buoyancy, respectively, $Y_M$ is the contribution of the fluctuation dilatation in compressible turbulence to the overall dissipation rate, $S_k$ and $S_\varepsilon$ are user-defined source terms, and $C_{1\varepsilon}$, $C_{2\varepsilon}$ and $C_{3\varepsilon}$ are model constants. $\mu_t$ is the turbulent viscosity, this can be defined as $\mu_t = \rho C_\mu k^2/\varepsilon$. The model constants are $C_{1\varepsilon} = 1.44$, $C_{2\varepsilon} = 1.92$, $C_{3\varepsilon} = 0.96$, $C_\mu = 0.09$, $\sigma_k = 1.0$, and $\sigma_\varepsilon = 1.3$.

Figure 7a–d show the pressure distributions for discharge valve opening lifts of 1 mm, 3 mm, 5 mm, and 7 mm. Owing to the sharp decrease of the through-area during the valve opening, the pressure drop was relatively large at approximately 0.1 MPa, as shown in Figure 7a. The pressure drop decreases to 0.01 MPa with an increase in the opening lift, as shown in Figure 7d. The pressure distribution in the overall flow channel was relatively smooth and symmetrical owing to the wing-guided structure of the valve core. Figure 8a–d show the velocity distributions for discharge valve opening lifts of 1 mm, 3 mm, 5 mm, and 7 mm. The flow velocity decreased as the opening lift increased from 15.08 m/s to 5.96 m/s. No bubbles were found in the CFD results. This differed from the flow visualization test findings.

This may be because of the complexity of the test suction conditions, for example, various local and along-the-way pressure losses of the suction pipes, as reported by Dong et al. [10].

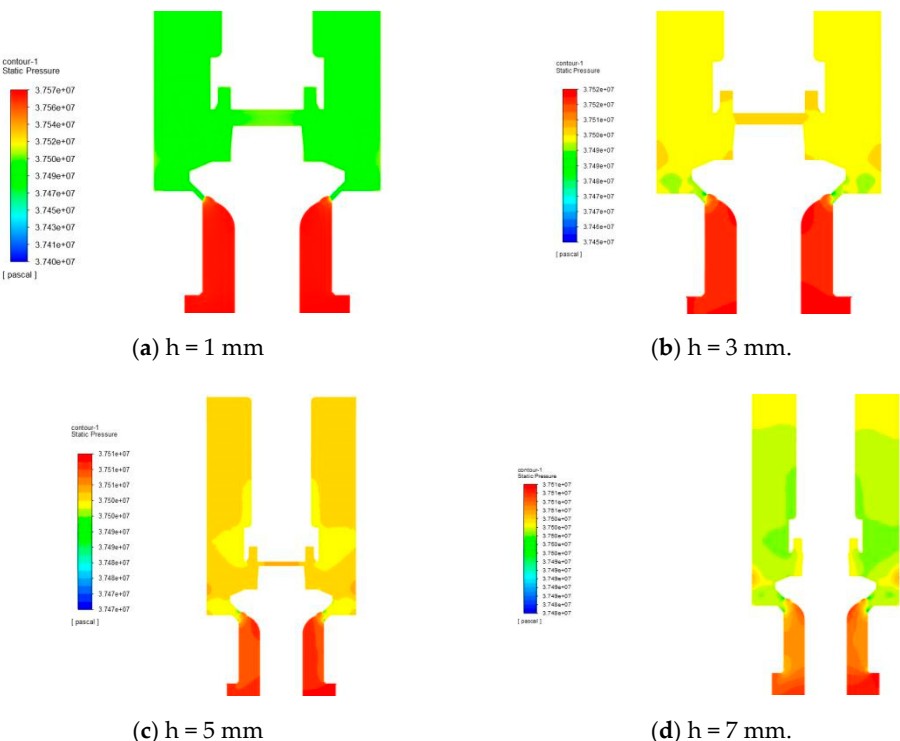

(**a**) h = 1 mm      (**b**) h = 3 mm.

(**c**) h = 5 mm      (**d**) h = 7 mm.

**Figure 7.** Pressure distributions for the different discharge valve opening lifts.

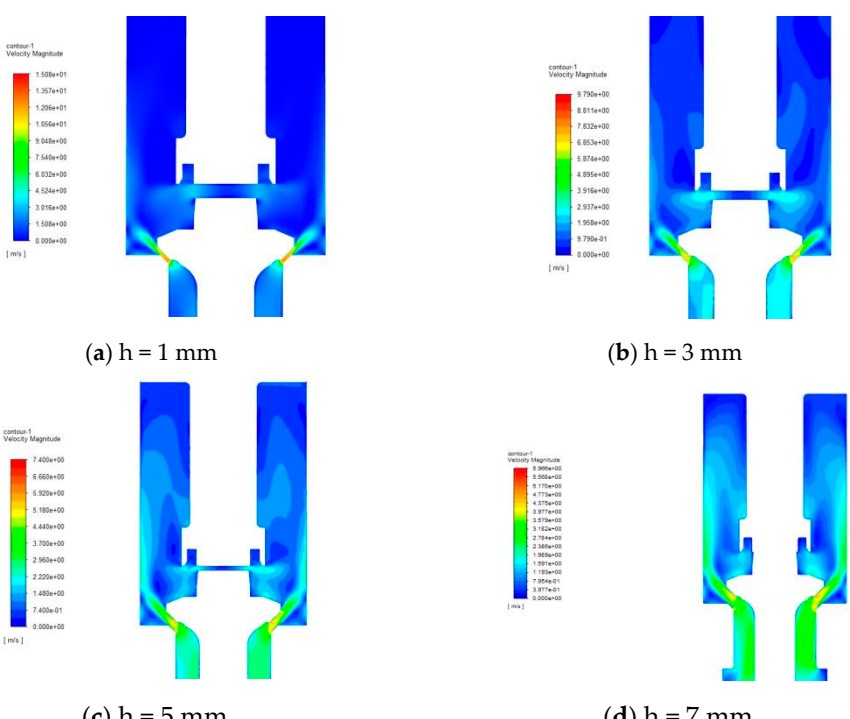

(**a**) h = 1 mm      (**b**) h = 3 mm

(**c**) h = 5 mm      (**d**) h = 7 mm

**Figure 8.** Velocity distributions for the different discharge valve opening lifts.

## 4. Results and Discussion

### 4.1. Comparison between Theoretical Analyses and Test Results for Valve Motion

The second-order nonlinear ordinary differential equation for characterizing the dynamic characteristics of the pump valve, known as the U-Adolph theory, is expressed in Equation (5).

$$T^2 h^2 h'' + h^3 + A h^2 - \varepsilon B f^2 + \varepsilon C f h' - \varepsilon D h'^2 = 0, \tag{5}$$

$$T^2 = \frac{m}{c} \quad A = \frac{G + F'_S}{c} \quad B = \frac{\varphi \xi \rho A^2 A_v r^2 \omega^2}{2 c l_v^2 \sin^2 \alpha} \quad C = \frac{\varphi \xi \rho A A_v^2 r \omega}{c l_v^2 \sin^2 \alpha} \quad D = \frac{\varphi \xi \rho A_v^3}{2 c l_v^2 \sin^2 \alpha} \tag{6}$$

where $m$ is the valve mass, $c$ is the spring stiffness coefficient, $F_S'$ is the spring preload, $l_v$ is the peripheral length of the valve gap, $A_v$ is the cross-sectional area of the pump valve closing, $\xi$ is equal to the reciprocal valve of $\mu^2$ ($\mu$ is the flow coefficient of the valve), $\rho$ is the density of the working medium, $\alpha$ is the complementary angle of the bevel angle, and $A$, $r$, and $\omega$ are the pump parameters.

For the BRW630/40 pump, the parameters were as follows: plunger diameter = 0.06 m; crank angular velocity = 69.43 rad/s; valve spring preload = 47.9 N; and density of working medium = 1000 kg/m$^3$.

MATLAB was used in conjunction with the Runge–Kutta method to solve the U-Adolph equation iteratively. This combined method assumes that the valve core opens when the crack angle is 25° and ignores the valve core closing singularity. The solution was obtained and reported in detail in a previous study [20]. The displacement of the pump valve was obtained, as shown in Figure 9. The maximum displacement was approximately 8.4 mm, which agreed well with the valve motion test result of 8.3 mm. Without considering valve closing hysteresis for the U-Adolph theory, the valve opening and closing durations were approximately 7.5 ms and 25 ms, respectively. The valve vibration phenomenon during closing and the valve opening and closing durations obtained from the U-Adolph prediction were inconsistent with the valve motion test results.

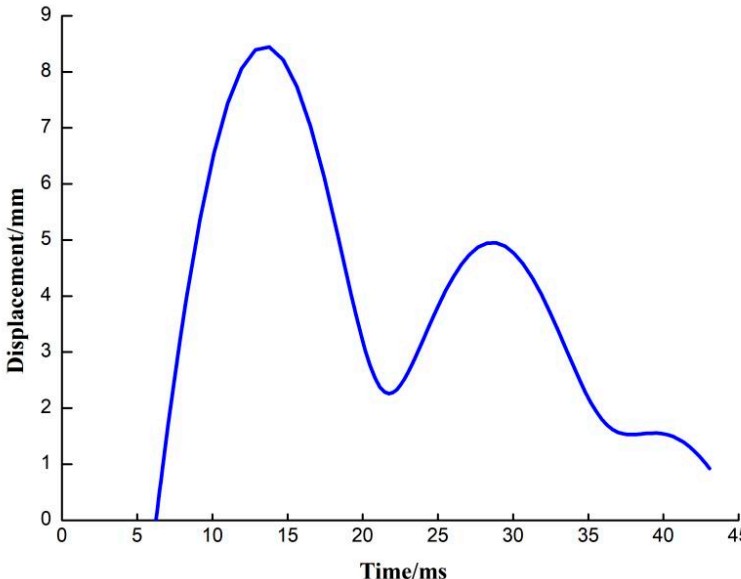

**Figure 9.** Displacement of the pump valve obtained from the U-Adolph prediction.

### 4.2. Visualization Results for Valve Opening and Closing Dynamic Characteristics, and Valve Opening Flow Characteristics

A sequence of images for the valve opening in the plexiglass liquid end of the reciprocating plunger pump is shown in Figure 10. At the beginning of the valve opening process, the flow was relatively smooth and the number of bubbles was not significant, as shown in Figure 10a. When the valve reached the maximum opening displacement, numerous bubbles were generated, as shown in Figure 10b. The bubbles continued to be generated

because of the valve-closing hysteresis, as shown in Figure 10c, and collapsed during the valve-closing process, as shown in Figure 10d. Bubbles were generated and gathered in the centerline of the valve. No contact with the wall was observed, which was likely to result in the absence of cavitation failure.

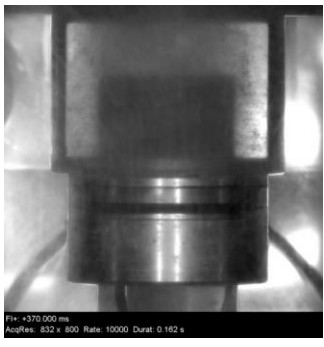

(**a**) Valve start opening

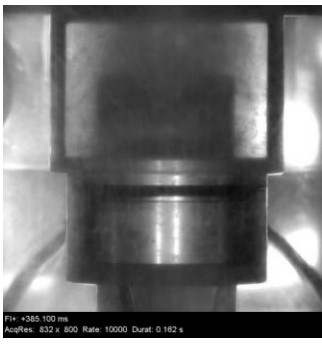

(**c**) Bubbles generating

(**b**) Valve maximum displacement

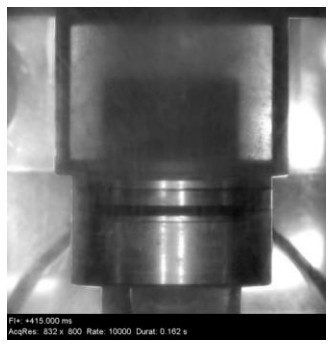

(**d**) Valve closing

**Figure 10.** Sequence of images for valve opening in the plexiglass liquid end of the reciprocating plunger pump.

*4.3. Error Analyses by Comparing the U-Adolph Predictions and the Valve Motion and Flow Visualization Test Results*

Table 2 shows the comparison between the U-Adolph predictions and the valve motion and flow visualization test results for error analysis. For the maximum displacement, the U-Adolph prediction is consistent with the valve motion test result. However, the maximum displacement for the prototype visualization test was larger than that of the other results. This overestimation may be related to the accuracy of the coordinate information obtained from the grid display and the change in the valve seat retention type. The valve opening duration results obtained from the U-Adolph theory and the valve motion test are consistent and are significantly smaller than those of the valve motion test, which most likely is limited by the high-frequency response incapability of the LVDT sensor [25–33]. The valve closing hysteresis phenomenon can be observed in Table 2 based on the valve motion and flow visualization test results, which differ from the U-Adolph prediction due to the valve core closing singularity, as also reported by Pei et al. [16].

**Table 2.** Comparison between the U-Adolph predictions and the valve motion and flow visualization test results.

| Parameter | U-Adolph Theory | Valve Motion Test | Flow Visualization Test |
|---|---|---|---|
| Maximum displacement (mm) | 8.4 | 8.3 | 11.4 |
| Valve opening duration (ms) | 7.5 | 29 | 9.5 |
| Valve closing duration (ms) | 25 | 38 | 35.5 |

*4.4. Effects of the Valve Opening Lift on Pressure, Velocity, and Discharge Coefficient Based on CFD Results*

The discharge coefficient $C_d$ was used to describe the effect of the valve opening lift on the pressure drop in the CFD results. The equation given in a previous study [34] is given by Equation (7).

$$C_d = \sqrt{\frac{\rho}{2\Delta p}} \frac{A_{in} u_{in}}{A(h)} \tag{7}$$

where $A_{in}$ is the inlet area, $\Delta p$ is the pressure drop, and $A(h)$ is the valve opening area, which can be calculated using Bussien's equation [35], given by $A(h) = \pi h \cos \theta D_{outer}$.

Figure 11 shows the discharge coefficients obtained from the CFD results. The results varied from 25.38 to approximately 11.40 (with an increase in the valve opening lift). The discharge coefficient remained constant once the valve opening lift reached approximately 5 mm. Wahono et al. [36] presented a similar conclusion by investigating the discharge coefficients measured in steady-state flow bench experiments and simulations for various valve lifts of a small motorcycle engine.

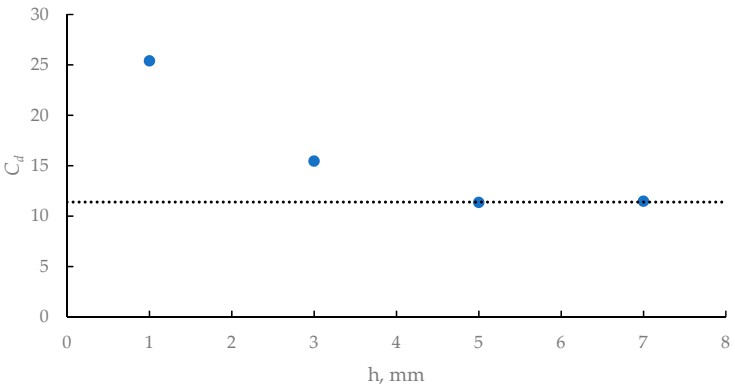

**Figure 11.** Discharge coefficients for different valve opening lifts.

*4.5. Research Application*

The method developed in this study was applied to develop a new horizontal quintuple single-acting reciprocating pump, with a rated flow rate of 1250 L/min and a rated pressure of 40 MPa. Figure 12a shows the CFD simulation of the velocity distributions for discharge valve opening lifts of 1 mm, 2 mm, and 12.5 mm. The flow velocity decreased as the opening lift increased from 14.4 m/s to 3.82 m/s, presenting reasonable flow characteristics and indicating good performance and high reliability. The newly developed reciprocating pump has been used as the coalmine emulsion pump in the Shendong mining area of the Buertai coalmine, China, since September 2021 [37], as shown in Figure 13. It delivers high-pressure media to the fully mechanized mining hydraulic support and is an important source for the continuous and efficient mining of the entire working face. Underground test results showed that the pump meets the high performance and reliability requirements of the ten million t/a coalmine.

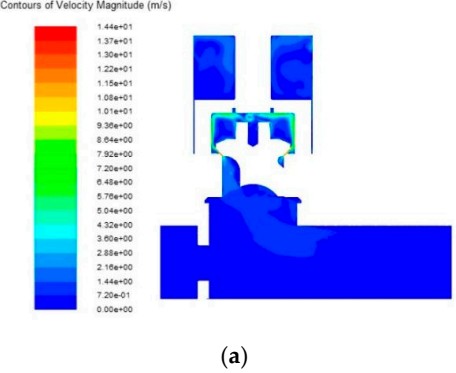

**(a)**

**Figure 12.** *Cont.*

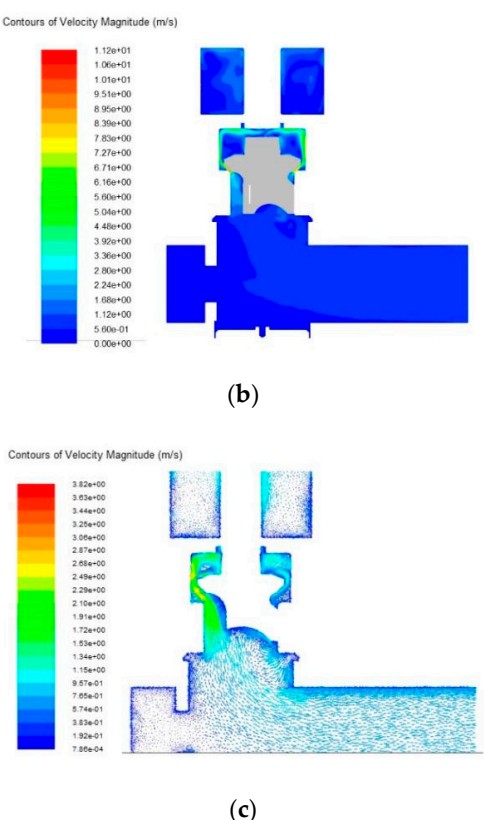

(**b**)

(**c**)

**Figure 12.** Velocity distribution of the newly developed reciprocating pump for different discharge valve opening lifts: (**a**) 1 mm; (**b**) 2 mm; and (**c**) 12.5 mm.

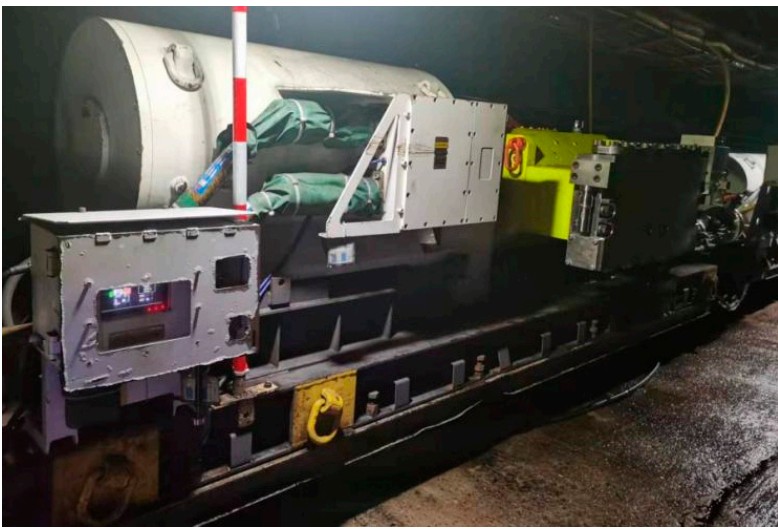

**Figure 13.** Underground test setup for the newly developed reciprocating pump.

## 5. Conclusions

The dynamic and flow characteristics of a quintuple reciprocating pump valve were experimentally investigated through valve motion and flow visualization tests. Three-dimensional CFD numerical analyses were performed in the present study. The results obtained from the two tests were mutually verified and compared with the U-Adolph theoretical prediction and CFD numerical results. The concluding remarks are as follows:

(1) The maximum valve opening displacement obtained from the valve motion test was approximately 8.3 mm, which agreed with the U-Adolph prediction. However, the

maximum displacement based on the grid display image analysis method for the flow visualization test was overpredicted at 11.4 mm.

(2) The valve opening duration obtained from the flow visualization test was consistent with the U-Adolph prediction at approximately 9.5 ms. The valve closing hysteresis phenomenon was clearly observed. The valve closing duration obtained from the flow visualization test was 42% higher than that predicted by U-Adolph. The valve opening and closing durations obtained from the valve motion tests were approximately twice as large as those obtained from the flow visualization test and the U-Adolph prediction, which were likely limited by the high-frequency response incapability of the LVDT sensor.

(3) Based on the flow visualization test, the flow characteristics were clarified as follows: during the valve opening process, the flow was relatively smooth; during the valve closing hysteresis, many bubbles were generated; and during the valve closing process, the bubbles collapsed. Bubbles gathered in the centerline of the valve and had no contact with the wall, which was expected to decrease cavitation failure.

(4) The valve opening process was numerically analyzed using the commercial code ANSYS-FLUENT. The CFD results show that the flow was relatively smooth, similar to the flow visualization test results. However, no bubbles were found in the CFD results, which may be related to the various suction pressure losses in the flow visualization test. The pressure dropped and velocity decreased as the discharge valve opening increased from 1 mm to approximately 5 mm.

(5) A new horizontal quintuple single-acting reciprocating pump with a rated flow rate of 1250 L/min and a rated pressure of 40 MPa was developed using the method presented in this paper. CFD results show that the velocity decreases from 14.4 m/s to 3.82 m/s with an increase in the discharge valve opening lift from 1 mm to 12.5 mm. Underground test results showed that the pump meets the high performance and reliability requirements of a ten million t/a coalmine in China.

**Author Contributions:** Conceptualization, R.L.; methodology, R.L.; formal analysis, W.W. (Wenshu Wei); data curation, D.W., H.L., J.Y., S.L. and W.W. (Wei Wang); writing—original draft preparation, R.L.; writing—review and editing, H.L., supervision, R.L.; project administration, R.L.; funding acquisition, R.L. All authors have read and agreed to the published version of the manuscript.

**Funding:** This research was funded by [Department of Policies, Laws and Regulations, MIIT, China] grant number [2011-14] And [CCTEG project] grant number [2020-TD-ZD015].

**Conflicts of Interest:** The authors declare no conflict of interest.

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
