# Peer review of "Experimental and Numerical Study on the Dynamic and Flow Characteristics of a Reciprocating Pump Valve"

_processes, doi:10.3390/pr10071328_

Round 1

Reviewer 1 Report

Li et al. presented "Experimental and numerical study on dynamic and flow characteristics of a reciprocating pump valve". In this work, flow visualization tests and computational fluid dynamics were applied on a wing-guided bevel discharge valve in a horizontal quintuple single-acting reciprocating pump.  They found that the maximum amount of valve displacement was 8.3 mm, and opening and closing duration for the valve were reported as 29 ms and 38 ms.  The maximum opening displacement of 8.3 mm was in agreement with U-Adolph prediction. It was also reported that the flow was relatively smooth when the valve was opening, and during the valve closing hysteresis, air bubbles were appearing. When the valve was closing, the bubbles were found to be collapsed.  Overall, the article is well-written and the data support the claims of the authors. I have the following minor comments.

- Authors should mentioned the limitations/shortcomings/ unanswered questions of the works reported in the literature in addition to mentioning their work. 

-What is the resolution of the LVDT sensor?

-What is the error in valve displacement and opening and closing durations?

Reviewer 2 Report

Methods lack error method description thus not trustworthy.

Please add an error analysis of Your results. Please compare the regression of Your model with other published ones.

Please add it to Your results.

Reviewer 3 Report

  1. The main finding and its significance should be highlighted in the abstract as well as in the conclusion.
  2. The differences of this study with others should be discussed.
  3. The literature review should be updated by including recent studies.
  4. Should cite recent papers published in Processes, to show that this paper is within the scope of the journal.
  5. The discussion should include some examples and the real applications of this study.
  6. The author should check the manuscript thoroughly, and make sure it free from any errors.
  7. The title is about experimental and numerical study. However, I cannot see the later contribution. Should include and discuss the method used to solve the system of  Eqs. (1)-(4), not only present the results.
  8. What are the boundary conditions for Eqs. (1)-(4)? Can this system be solved numerically without boundary conditions?

Round 2

Reviewer 2 Report

The authors responded correctly to my comments. The manuscript can be published in the present form

Reviewer 3 Report

The manuscript has been improved as suggested.